# Current velocity, water quality, and benthic taxa as predictors for coral recruitment rates on the Great Barrier Reef

**Matilde A. Drake**[1]*, **Sam H. C. Noonan**[1], **Mariana Alvarez-Noriega**[1], **Ahmad R. Rashid**[1,2], **Katharina E. Fabricius**[1]

**1** Australian Institute of Marine Science, Townsville, Australia, **2** James Cook University, Townsville, Australia

* matilde.drake@outlook.com

## Abstract

Coral reefs worldwide are experiencing frequent disturbances, rendering coral recruitment critical for population recovery. This large-scale study identifies environmental, spatial, and biotic drivers of coral recruit densities at 141 stations stratified across seven regions and three depths (1, 5, and 15 m depths) with contrasting environmental conditions across and along the Great Barrier Reef and the Torres Strait. Settlement tiles were deployed for two years, with coral densities and benthic cover quantified following retrieval. Benthic communities were assessed from tile images using the point-classification AI program ReefCloud. Environmental data were derived from *in situ* readings and environmental models. Across all sites, coral recruit densities averaged 187 ± 12 m$^{-2}$ (SE), with region-wide averages ranging from 43.5 ± 12 m$^{-2}$ to 247 ± 32 m$^{-2}$. Mean densities were 3-fold higher in the four clear-water regions compared to the three turbid-water regions. Boosted regression tree analyses showed that densities declined with increasing current velocity, sedimentation, and depth, and increased with increasing pH. From lowest to highest observed levels of current velocity, recruit densities declined by ~530 m$^{-2}$. From lowest to highest sedimentation, densities declined by ~300 recruits m$^{-2}$. Even relatively minor increases in sediment deposits from 0.1 to 38 mg cm$^{-2}$ were associated with a monotonic decline of ~130 recruits m$^{-2}$. Recruit densities were also weakly positively related to the cover of turf and crustose coralline algae on tile tops, and negatively related to fleshy invertebrate cover on the tile undersides. Some variation in the cover of these benthic taxa was also related to environmental conditions (e.g., sedimentation and currents), suggesting the possibility of additional indirect environmental effects on recruit densities. Our results highlight the strong role of current velocity and water quality as regulators of coral recruitment success, likely influencing the capacity of reef sites to recover after a disturbance.

## Introduction

Coral reefs are the Earth's most diverse marine ecosystems, serving as a critical habitat and nursery to thousands of species while providing socioeconomic benefits to millions of people through, for instance, cultural heritage, fisheries, tourism, and protection against coastal

**Data availability statement:** The data underlying the results presented in the study have been uploaded to the public repository figshare and are made accessible under the following link: https://doi.org/10.6084/m9.figshare.26662129

**Funding:** This research was funded by the Reef Restoration and Adaptation Program, a partnership between the Australian Governments Reef Trust and the Great Barrier Reef Foundation (https://gbrrestoration.org/), and the Australian Institute of Marine Science (https://www.aims.gov.au/). The funders had no role in study design, data collection and analysis, decision to publish, or preparation of the manuscript.

erosion [1]. These vital ecosystems face severe threats due to anthropogenic climate change and ocean acidification, water pollution, and unsustainable fisheries [2]. Climate change has exacerbated the intensity and frequency of disturbances, such as cyclones, marine heat waves, mass bleaching, disease, and crown of thorns outbreaks, quashing coral populations [2,3]. The recovery of coral populations is in part reliant on the settlement and post-settlement survival (i.e., recruitment) of coral recruits [4–9]. Given the important role of coral recruitment success for ecosystem models on reef dynamics, empirical data are needed to quantify the effects of environmental and biotic factors on recruit densities at intra-reefal to regional scales. Robust empirical data and models can also assist in identifying areas of natural reef recovery potential, and optimize spatial prioritization of restoration and management strategies.

Many different environmental factors can influence coral settlement and recruitment rates. Light availability, for one, is an important factor determining coral settlement choices and post-settlement survival. A delicate balance exists, as adequate light is essential for the photosynthetic capabilities of algal symbionts [10], yet light-exposed surfaces are also areas of fast algal growth and are heavily grazed [11]. In light-saturated shallow waters (0-10 m), recruits therefore tend to settle in cryptic non-exposed areas [10,12,13], while at greater depth, larvae mostly attach to light-exposed surfaces [10,12,13]. Recruitment on cryptic and exposed habitats vary across differing environments, such as turbid and clear water areas, as coral larvae also preferentially chose cryptic surfaces over those that are sediment-laden [14]. Recruitment rates and choices in orientation are therefore not only a result of light availability, but also of water quality (including sedimentation), grazing pressure, and competition with other organisms such as algae.

The role of hydrodynamics in coral recruitment success and juvenile distribution is still poorly understood. Hydrodynamics are primarily governed by local winds, tides, and large current inflows and are highly variable within and among reefs [15,16]. Water velocity influences population connectivity through larval dispersal within reefs, to nearby reefs, or even past habitable zones [17]. Currents can reduce or enhance larval retention (self-seeding) on the reef, as strong currents can either transport larvae away from the reef or trap them by creating eddies in their wake [18]. Strong currents can also prevent the ciliated near-passively floating larvae from landing and settling on the reef, or by repeatedly overturning coral rubble as settlement substratum, thus increasing mortality [19,20]. Water flow improves coral metabolism by breaking down the coralline boundary layer and enhancing particle interception, hence post-recruitment, corals in strong currents can show higher growth and resilience to environmental stressors including to marine heat waves [21,22].

Sedimentation and nutrients can also impact many aspects of coral recruitment [23]. Off the Northeast Australian coast, the degradation of water quality, especially on reefs near the coast, is primarily caused by soil erosion from cattle grazing lands and fertilizer application to sugarcane cultivation [24–26]. In suspension, sediments promote macroalgae and filter feeders and reduce light reaching the seafloor, hindering overall reef growth [27]. Once settled, sediments may also disrupt the attachment of coral larvae by physical obstruction or by concealing the biotic cues of the substrata [28]. If larvae successfully attach, sediments may smother recruits, reducing light availability and gas exchange. Even small rises in sediment levels can have adverse impacts on coral settlement [29]. Elevated concentrations of nutrients can also pose deleterious effects on recruitment, as a fertilization bottleneck, or indirectly by enhancing algal competitors [30–32]. Another water quality aspect is ocean acidification, from increasing atmospheric $CO_2$ and high respiration in high-nutrient environments (coastal acidification), which reduces coral recruitment through complex pathways [33,34].

The composition of the benthic community also affects coral recruitment rates. Almost all reef surfaces are occupied [35], and while increased space availability facilitates coral

recruitment [36], the presence of certain taxa can promote or hinder it. Encrusting species such as some crustose coralline algae (CCA), sponges, or bryozoans, can overgrow coral recruits [11,28,37]. Some species of CCA, such as *Neogoniolithon fosliei,* rapidly slough off their outer layer as an anti-fouling defense mechanism, excluding corals from attachment [28]. Biochemical interactions play a role, as certain macroalgal and sponge species release allelopathic compounds that deter, and can reduce the presence of, nearby coral recruits [38]. The three-dimensional matrix of algal turfs can trap sediment, even in low sediment environments, thereby deterring or smothering recruits [39]. Fast-growing species, such as macroalgae, may overgrow or outcompete recruits for light resources [40]. In low light conditions the growth of some algal groups (*Peyssonnelia* spp.) is promoted, which can negatively affect coral juvenile survival [37]. Competition also occurs between corals; for instance, established scleractinians present a "wall of mouths," intercepting coral larvae in the water column [41], and some corals can eliminate nearby recruits via "sweeper" tentacles [42]. While many interactions among coral recruits and benthic communities are known, knowledge stems largely from clear water systems, leaving competitive recruit interactions in turbid water systems understudied.

Coral settlement can conversely be promoted by certain benthic organisms. Certain CCA species including *Titanoderma* promote recruitment for many coral species, especially those that shed their outer layers slowly and those with biofilms that induce settlement via biochemical cues [28,43]. Moreover, shedding CCA indirectly benefits adjacent recruits through the removal of organisms that may otherwise outcompete them [44]. Coral recruitment may increase in early successional phases following a disturbance. Bare substratum, biofilms, and preferred biotic taxa become available to coral recruits, presenting them with a narrow "recruitment window" where the probability of survival is greater [45]. The presence of adult corals and conspecific recruits settling in clumps can also increase coral recruitment [46,47].

In this large-scale study, we investigated the main predictors of coral recruitment rates along the 2000 km long Great Barrier Reef (GBR) and adjacent Torres Straits. To do so, we explored the direct, indirect, and interactive effects of environmental conditions and key benthic taxonomic groups on coral recruitment across various spatial scales. We quantified variations in coral recruit densities and taxonomic richness in relation to their environmental conditions (including current velocity and water quality) and biotic communities on settlement tiles that had been deployed for two years across a broad range of reef habitat types. We also examined the composition of benthic communities on the tiles and their influence on coral recruit densities. Our empirical data serve to parameterize reef ecosystem models and inform conservation practitioners about natural recruit densities in different reef environments, contributing to predicting areas of high or low inherent reef resilience.

## Methods

### Ethics statement

This research was conducted in compliance with the Great Barrier Reef Marine Park Authority (GBRMPA); fieldwork and sample collection granted under permit G21/44774.1. Animal Ethics Committee approval was not required, as targeted and collected samples excluded 'animals,' i.e., non-human vertebrates, cephalopods, and crustaceans. No protected species were sampled.

### Study sites

This study took place at twelve reefs off the Queensland coast, stretching from the Torres Strait (Latitude 9.7°S) to the Capricorn Bunker reefs (Latitude 23.9°S) in the GBR (Fig 1).

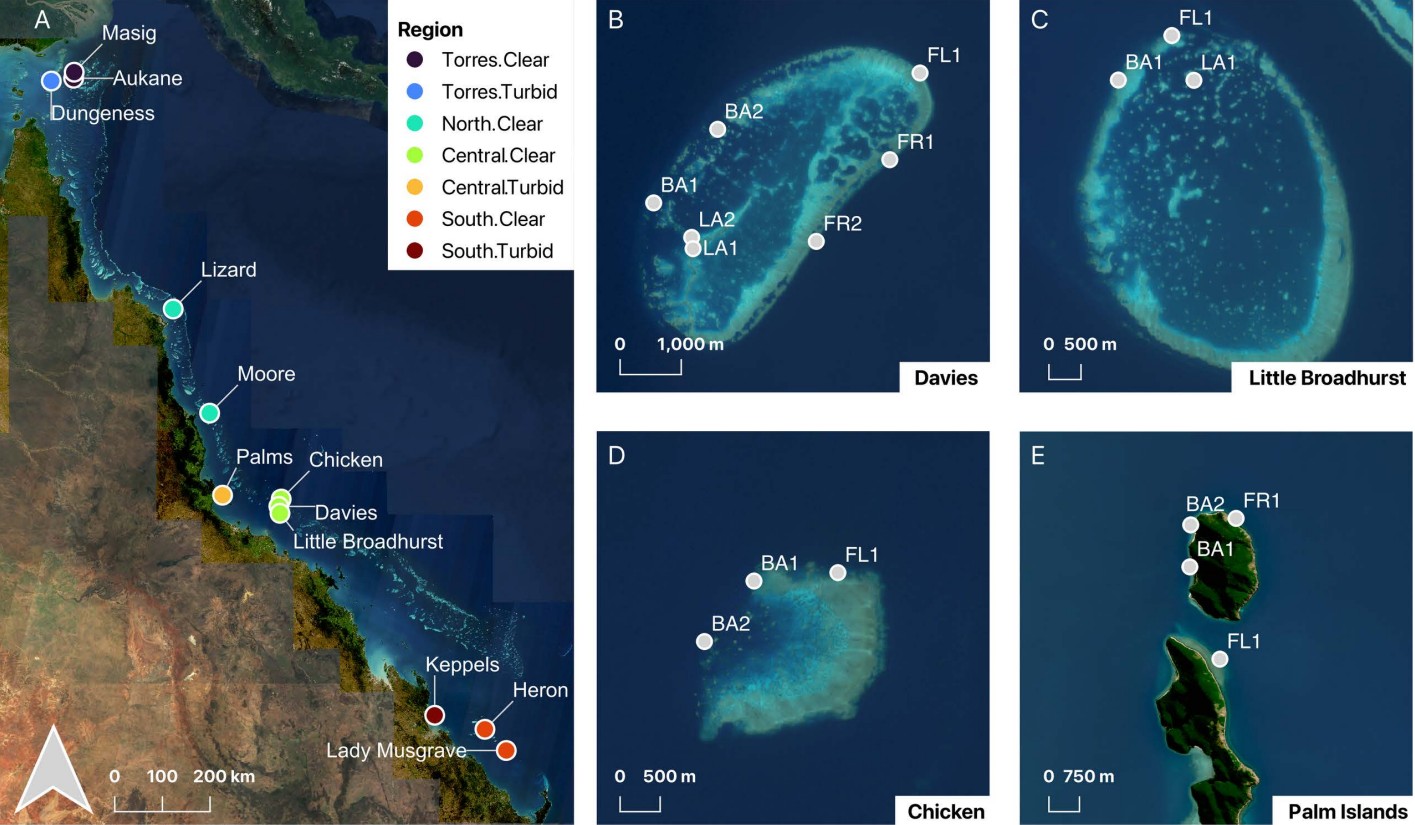

**Fig 1. The study reefs and regions in the Great Barrier Reef and Torres Strait, NE Australian coast** (A). For illustration, the within-reef sampling structure is shown for the reefs in the regions Central.Clear (B-D: Davies Reef, Little Broadhurst Reef, Chicken Reef) and Central.Turbid (E: Palm Islands). Color gradient points indicate reef locations and their respective regions. Grey points represent site locations (FR = front reef; FL = flank reef; BA = back reef; LA = lagoon). Base map (A): republished from Sentinel-2 cloudless 2016 [50] (contains modified Copernicus Sentinel data from 2016 and 2017) under a CC BY license, with permission from EOX IT Services GmbH [51], original copyright 2017. Reefs (A-E): republished from [52] (contains modified Copernicus Sentinel data from 2015-2024) under a CC BY license, with permission from AIMS, original copyright 2024.

These reefs were a priori categorized into seven regions based on their latitude, proximity to mainland Australia, and subsequent water clarity: Torres.Clear, Torres.Turbid, North.Clear, Central.Clear, Central.Turbid, South.Clear, and South.Turbid (North.Turbid was not possible due to crocodiles; S1 Table). Due to their similar formation and sea-level histories, Torres Strait and GBR reefs contain geomorphic and zoogeographic similarities [48]. As a result of their remoteness, reefs of the Torres Strait remain largely understudied in comparison to their well-documented neighbors in the GBR [48,49].

Within these regions, we established a total of 141 monitoring stations at 63 sites around the reef perimeters, with a complex nested sampling design to reflect a broad range of reef habitat types (Fig 1, S1 Table). Sites within reefs were classified by their exposure to wave energy and geomorphology as reef front, flank, back, and lagoon (in descending order of relative wave exposure). The front, flank, and back sites line the exterior rim of the platform reefs and island fringing reefs, while the lagoon is a sheltered habitat on the interior portion of some platform reefs. Front reef sites face the prevailing winds–bearing the highest exposure to waves and storm energy. Back reefs are the least exposed exterior sites, facing the leeward side. Flank reefs lie in-between front and back reefs. Each site contained up to three depth levels, namely at ~ 1 meter (m) depth (reef flat), 5 m (reef crest or upper shallow slope), and 15 m (deep slope) from the surface at mean tide. Every site included a reef flat

and a shallow slope, but only one reef per region included deep slopes (except for the South. Turbid, which was too shallow).

## Settlement tiles

A total of 710 individually tagged settlement tiles, comprised of 11.5 x 11.5 x 0.3 cm sandpaper-roughened polyvinyl chloride (PVC), were placed across the sites in 2021 (January, April, May, October, for Central GBR, Torres Strait, Southern GBR, and Northern GBR, respectively). In comparison to other materials (such as terracotta, ceramic, limestone, and glass), the percent coverage of certain benthic taxa settling on PVC has been shown to more closely reflect surrounding reef substrata [53]. At each site and depth level, five replicate tiles were deployed approximately 1–5 m distance between each other, in a cluster, and fixed on firm substratum. Each tile was horizontally fixed approximately 2 cm above the reef substrate on stainless-steel baseplates (following Mundy [54]); the space between the tile and substratum allowed for recruitment of cryptic taxa on the shaded bottom side, while light-exposed and grazed communities developed on the top side.

## Benthic community composition

After 20–24 months (precise time per reef in S1 Table), 622 of the original tiles were retrieved while 88 were lost due to dislodgement. Settlement tiles were carefully removed from their baseplates and placed into individual plastic bags while still underwater. Tiles were immediately returned to the research vessel and transferred from their bags into a flowthrough bin of *in situ* seawater, keeping them submerged at all times. Before transfer, the bags containing the tiles were gently agitated to dislodge all sediments for future examination.

Onboard the vessel, a dissection microscope was used to record the number and diameter of each coral recruit on the upper and lower tile sides and tile edges. Recruit size was measured as the largest diameter to the nearest 1 mm. Coral taxonomic identification, made by observers experienced with coral identification, using morphological features of the corallites, was recorded where possible, limited to genus and family level or unknown [55]. Subsequently, photographs of the top and bottom sides of fresh settlement tiles were captured for later analysis of the benthic community composition. Pictures were taken using a Nikon D300 camera and 60 mm NIKKOR lens linked with a set of two remote controlled Nikon SB-R200 strobes.

The light-exposed top sides and shaded bottom sides of the tiles were separately analyzed for their benthic community composition using a combination of evenly spaced and random point count methods [56]. To do so, the photographs, cropped to the tile edges, were imported into the software ReefCloud [57]. This software uses a class of deep learning artificial intelligence (Convolutional Neural Network) developed for coral reef benthos imagery analysis [58]. ReefCloud efficiently categorizes coral reef communities, with up to 80–90% accuracy [59]. The benthos under 50 points (20 evenly spaced and 30 random) per 132 cm² tile surface, were identified into label set categories of benthic taxa frequently observed on the tiles (defined in S3 Table). Across the dataset, 45% of the evenly spaced points (i.e., 18% of the entire dataset) were labeled by a single observer to train the AI: training points were distributed across all reefs and depths, targeting less common label set categories, to improve AI results. The remaining evenly spaced points and all random points (i.e., 82% of the dataset) were machine annotated. Point counts were converted into percent coverage data, after discarding "trash" points such as those on tile number tags. The (human annotated) training points replaced machine points for calculation of the percent coverage data. After image processing, label set categories were grouped into functional groups for statistical analyses (S3 Table). In the

preliminary analysis, changes to recruit density varied between bryozoa and other groups of fleshy invertebrates (ascidians and sponges), and thus byrozoa were kept separate from the 'fleshy invertebrates' functional group in all analyses.

To validate ReefCloud AI software benthic identification accuracy, a random subset of 25% of tile images uploaded to ReefCloud were chosen, and all 20 evenly spaced grid points were human-annotated. Confusion matrices are shown in S3 and S4 Fig, in which the "actual" human annotated points were plotted against the "predicted" machine (from the random subset of tile images) annotated points as a percentage of the actual count:

$$\frac{predicted\ point\ frequency}{actual\ point\ frequency} \cdot 100 = actual\%$$

Some taxa were not well represented by the point identification method. In particular, macroalgal thalli, for instance in *Sargassum* spp., sometimes fell outside the photo grid and were thus unaccounted for. In addition, for analyses outside the scope of this study, some macroalgae were too large and were removed from tiles before pictures were taken. As a result, macroalgae yielded low mean cover (0.61% ± 0.10 top; 2.42% ± 0.18 bottom). Hard coral recruits were too small to be adequately represented by percent cover metrics (>1%), hence counts and their individual diameter estimates were used.

The density of coral recruits was calculated as: recruit abundance summed across all tile surfaces (top, bottom, and sides), divided by upper tile surface (132.25 cm$^2$), and multiplied by 10,000 to upscale to 1 m$^2$ (similarly to Edmunds et al. [60]). This standardization of recruit densities over a 2D surface (i.e., not accounting for complexity) represents an upward biased estimate, while a presentation in 3D (i.e., division of densities by upper plus lower tile surface) would represent a downward biased estimate [60], however, the choice of standardization does not affect the correlations shown below.

## Environmental data

To observe the effect of settled sediments on coral recruits and the benthic taxa, we measured the dry weight of sediments retained within the bags that each tile was collected in. First, the seawater containing the sediments was passed through a 710 μm sieve to remove large particulates, before being filtered through pre-weighed glass microfiber filters (Whatman GF/F: diameter = 47 mm, nominal pore size = 0.7 μm) attached to a vacuum manifold. After suspended solids were concentrated on the filters, ~50 mL of freshwater was used to flush salt contents from the sediments. In the laboratory, sediment filters were dried at 50°C for a minimum of six days. Sediment dry weight was then determined by subtracting the initial filter weight from the final weight of the filter containing the sediments.

Long-term mean environmental conditions of each study site and depth were derived from the coupled three-dimensional hydrodynamic and biogeochemical model 'eReefs' [61]. Daily means were extracted from the 1st December 2010 to 1st December 2018, the longest available period for which model outputs were consistent. The hydrodynamic model GBR1 (version H2p0) is run at 1 km horizontal resolution in 10-min steps, driven by wind, atmospheric pressure gradients, surface heat and water fluxes, and open-boundary conditions such as tides and low frequency ocean currents as input variables [61]. The hydrodynamic model drives movement and concentration of nutrients and sediments within the GBR4 Biogeochemical model (version 3p1a) at 4 km resolution. These nutrients and sediments are modeled by the effects of coastal runoff as well as biotic nutrient cycling mechanisms linked to plankton, seagrass, and coral. These models are built from wind, rainfall, pressure, air, and dew-point temperature data from the Bureau of Meteorology (BOM) (BOM's Access-R: [62]) [61]. The following

parameters were modeled: total nitrogen (TN) as proxy for nutrient exposure, total carbon (TC), total alkalinity (TA), aragonite saturation state, pH, Secchi depth (a measure of water clarity), salinity, and temperature. The time series of each modelled environmental variable was summarized by its mean. Additionally, estimates of mean horizontal water velocity at the sea bottom (Ubed mean, m s$^{-1}$) for each site and depth were extracted from available models that used the Simulating Waves Nearshore (SWAN) model, which computes wind-generated waves hourly at horizontal resolution based on long-term wind data observations as well as fine-scale bathymetry and fetch (~30 x 30 m; [63–65]). Since water velocity estimates were not available for the Torres Straits, we fitted a linear mixed effects model in a Bayesian frame-work to predict current velocity (Ubed mean) using the R package 'brms' [66]. An interaction between site and depth could not be used as an explanatory variable since only one lagoon site had a deep level. Instead, we included a combination of site and depth (e.g., BA_S for shallow back sites) as a single explanatory variable, and we excluded the Ubed mean estimate from the deep lagoon environment from this model. The model also included a random effect of reef, and it had an R$^2$ value of 0.53 (95% credible interval: 0.43–0.61). Using this model, we predicted the expected Ubed mean for the Torres Strait sites at each depth. The spatial scale of each environmental predictor can be found in S2 Table.

## Statistical analysis

All statistical analyses were conducted via the program R versions 3.4.2 and 4.1.1 [67] using the packages 'vegan' [68], 'abt' [69], 'ereefs' [70], 'brms' [66], and 'ggplot2' [71].

We calculated the difference in mean coral recruit taxonomic richness across turbid and clear regions, with and without the inclusion of tiles with zero recruits with a non-parametric Mann-Whitney test (α = 0.05).

The variance of recruit density per m$^2$ (calculated from bottom, edges, and tops of tiles combined) was examined in relation to the spatial, environmental, and benthic communities (predictors listed in S2 and S3 Tables). Non-parametric aggregated boosted regression tree analyses (ABT; R package 'abt' [69]) were utilized to ascertain the most influential predictors of recruit density. ABT models are built using machine learning methods, where algorithms learn the relationship between predictors and responses, based on ensembles of classification and regression trees. ABT can effectively handle numeric and categorical predictor variables measured across diverse scales. This type of analysis is particularly advantageous for models with complex, non-linear interactions as the model outcomes are robust against predictor transformations and outliers [69]. In response to the confounded nature of many of the pre-dictors (e.g., nutrients being highest in turbid inshore water, aragonite saturation state being a function of temperature, and crustose coralline algae being related to turbidity), two separate sets of analyses were conducted: (1) environmental predictors of coral recruit density and (2) benthic taxa on tiles as predictors of recruit density. Region and Depth were added to the environmental models, while Reef was not included as many of the environmental data were almost invariant within reefs (salinity, temperature, etc.). Site-type was also not included since the confounded current velocity (Ubed mean), a finer continuous variable, was of greater interest. The initial environmental recruit ABT included all nine environmental and the two spatial predictors, with interactions. Model predictors were first parsed down by dropping the least influential predictors allowing for 2-way interactions and adding monotonicity where appropriate [69]. Final ABTs were plotted as partial dependency plots, illustrating the effect of each predictor while all other predictor effects were held constant.

A redundancy analysis (RDA; R package 'vegan' [68]) was employed to visually assess associations among tile benthic communities, recruit density, and all spatial and

environmental conditions. Incomplete data (due to image loss or sediment weight loss) were removed, with 32 tiles being discarded for the RDA. Both the benthic communities and environmental vectors were scaled based on their eigenvalues. An ANOVA like permutation test (non-sequential partial model, 1000 iterations; anova.cca; R package 'vegan') was applied to the RDA results to assess the significance of each predictor in the presence of all other predictors [72].

## Results

After approximately two years of deployment, the top sides of the tiles closely resembled the surrounding reef substrata (S1 and S2 Figs, Fig 2). The top sides of tiles were algal dominated, with crustose coralline algae (CCA), turf, and *Peyssonnelia* spp., collectively covering on average 91.4% ± 0.42 (SE) of the top tile surfaces. Among these algal communities, an inverse relationship between mean CCA and turf cover was evident across the seven regions. Mean CCA cover increased offshore from north to south and was much lower on turbid than clear-water reefs (Figs 2 and 3). Communities also contrasted between the top and bottom sides of the tiles (Figs 2 and 3). Bottom communities predominantly featured fleshy filter-feeding invertebrates (including ascidians and sponges) (mean cover 31.4% ± 0.80), CCA (29.1% ± 0.73), and bryozoans (16.3% ± 0.59).

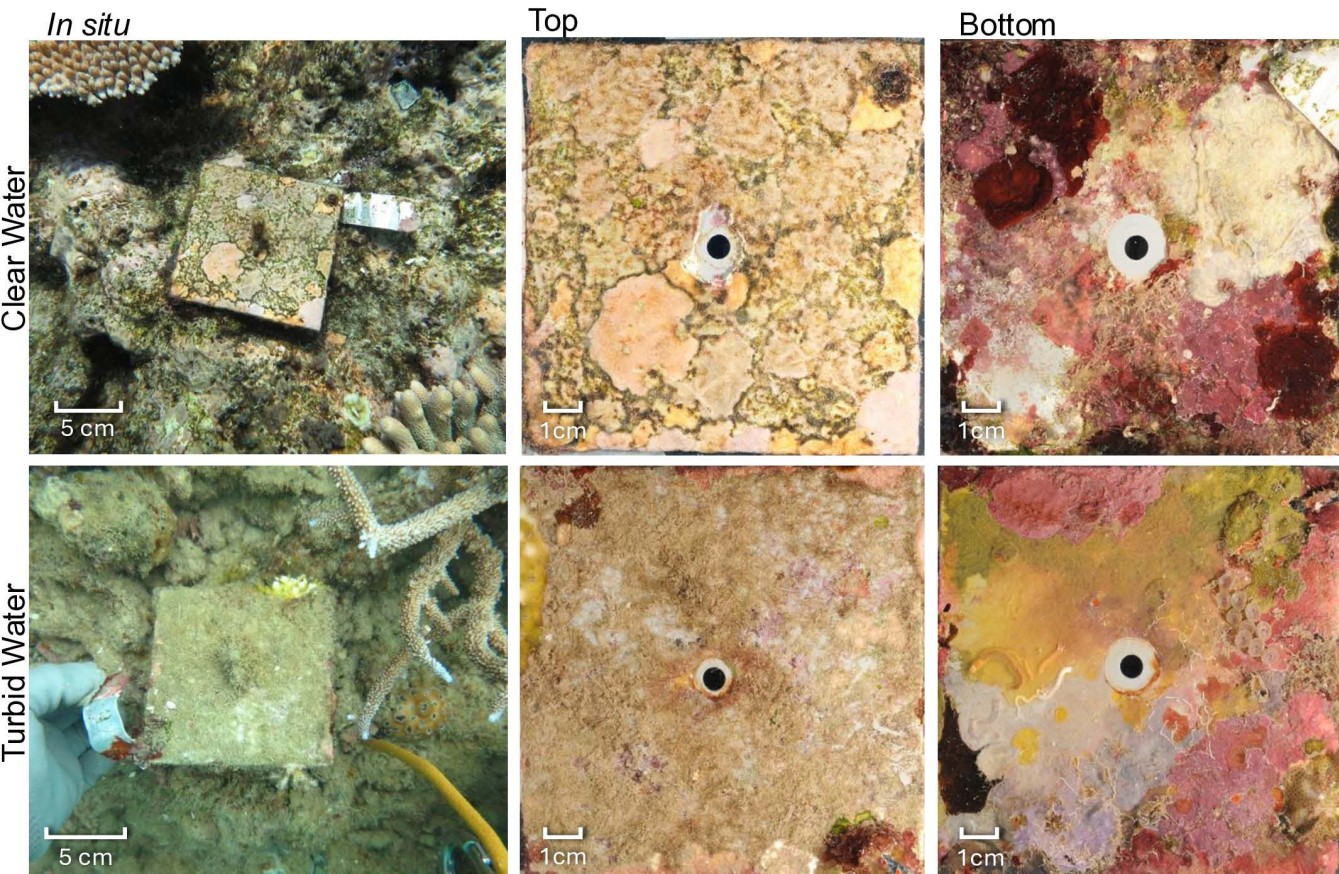

**Fig 2. Examples of tiles from a clear-water and a turbid-water site, *in situ*, and of their benthic communities on the tops and bottoms after sediments were removed (see S1 and S2 Figs).**

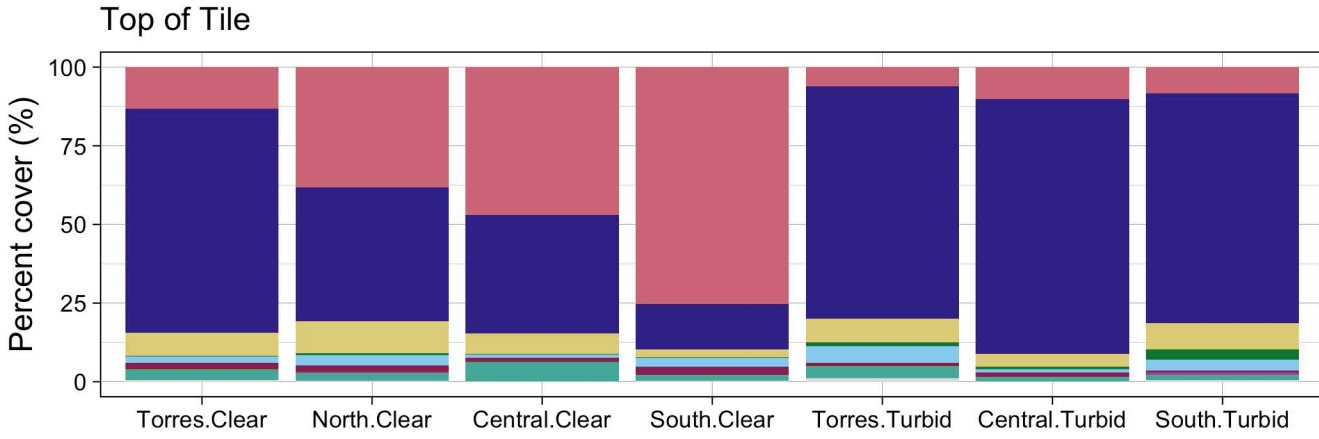

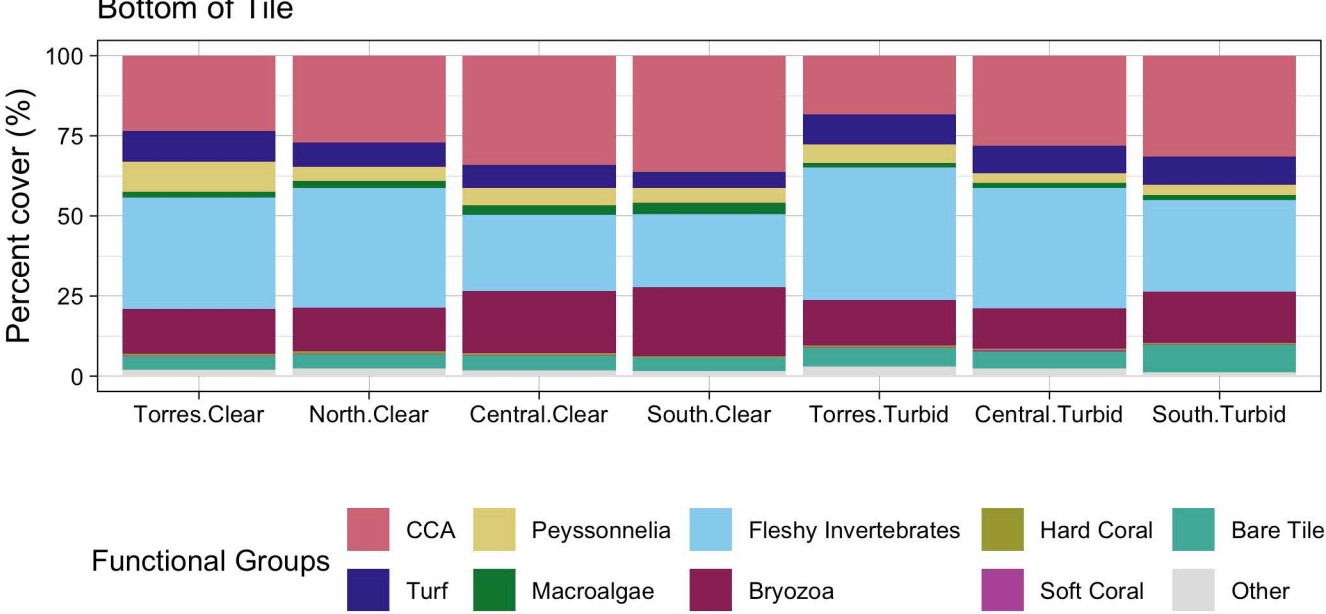

**Fig 3. Mean percent coverages of benthic taxa on the top and bottom sides of the settlement tiles in each of the seven GBR regions.**

In total, 1535 coral recruits were counted across the 622 tiles; 4% were recently dead. Recruit counts averaged 2.47 ± 0.16 (SE) per tile, with 36.8% of tiles having no coral recruits. The recruits were predominantly found on tile bottoms (83.8%), with 10.3% and 6.1% found on tile sides and tops, respectively. Extrapolating from the density of all tile surfaces to the reef would yield an estimated $187 \pm 12$ recruits $m^{-2}$ of reef (2D, i.e., disregarding surface complexity), or $93.3 \pm 6$ recruits $m^{-2}$ of total combined top and bottom surface area. *Acropora* (50.0%), *Pocilloporidae* (24.6%), and *Porites* (6.2%) were the most abundant taxa, while recruits of unknown identity constituted 16.9%. The diameter of the up to 24 months old recruits averaged $7.4 \pm 0.24$ mm (median of 3.0 mm). Models examining patterns in the number of recruits on the different tile surfaces separately (i.e., top, bottom, and sides) in relation to environmental variables and benthic communities were weak and uninterpretable, so recruit numbers were combined per tile over all surfaces for further analyses.

Total recruit density was highest at Little Broadhurst reef (564 ± 133 recruits m$^{-2}$ (SE)) in the Central.Clear region (where mean density was 247 ± 32 recruits m$^{-2}$) and Lizard Island (513 ± 66 recruits m$^{-2}$) in the North.Clear region (North.Clear mean density = 244 ± 35 recruits m$^{-2}$). The South.Turbid region at the Keppel Islands exhibited the lowest recruit density (mean = 43.5 ± 12 recruits m$^{-2}$). Overall, mean recruit density, was three times higher in clear-water regions (mean = 219 ± 14 recruits m$^{-2}$, median = 151 recruits m$^{-2}$) compared to turbid-water regions (mean = 74.0 ± 11 recruits m$^{-2}$, median = 0) (Fig 4). In all seven regions, *Acropora*, followed by *Pocillopora*, and *Porites* were most abundant, except for in the South. Turbid where no *Porites* were identified. Mean coral richness (number of taxa per tile) was also twice as high in clear-water compared to the turbid regions (1.23 ± 0.05 (SE) vs. 0.60 ± 0.07; Mann-Whitney test: W = 45140, p-value < 0.001), although richness is likely underestimated due to the presence of unknown taxa. The strength in the p-value was largely due to zero abundances; upon exclusion of zeros, the difference in richness was marginally significant (p-value = 0.035).

The environmental ABT model found coral recruit densities to be best predicted by three of the environmental and the two spatial parameters (Fig 5), although several alternative models were also relatively strong. Long-term mean current velocity was the strongest predictor (RI: 43.09%), and changes along the current gradient from 0.0062 to 0.57 m s$^{-1}$, were associated with a mean reduction of 530 recruits m$^{-2}$. The results also highlighted the role of water quality, with three of the predictors being water quality related. The four clear-water regions all had higher recruit densities than the three turbid-water regions (approximately 2-fold greater), all else held constant. These clear-water regions, and particularly the North. Clear and Central.Clear, were characterized by high Secchi depth, low sedimentation, and low total nitrogen. Sediment deposits, which ranged from 0.2 to 328 mg cm$^{-2}$ dry weight, were associated with a reduction of ~260 recruits per m$^2$. Differences between the highest and lowest observed pH from 8.05 to 8.11 were associated with 100 additional recruits per m$^2$. Across depth, total recruit density decreased by ~75 per m$^2$ from the reef flat (1 m depth) to

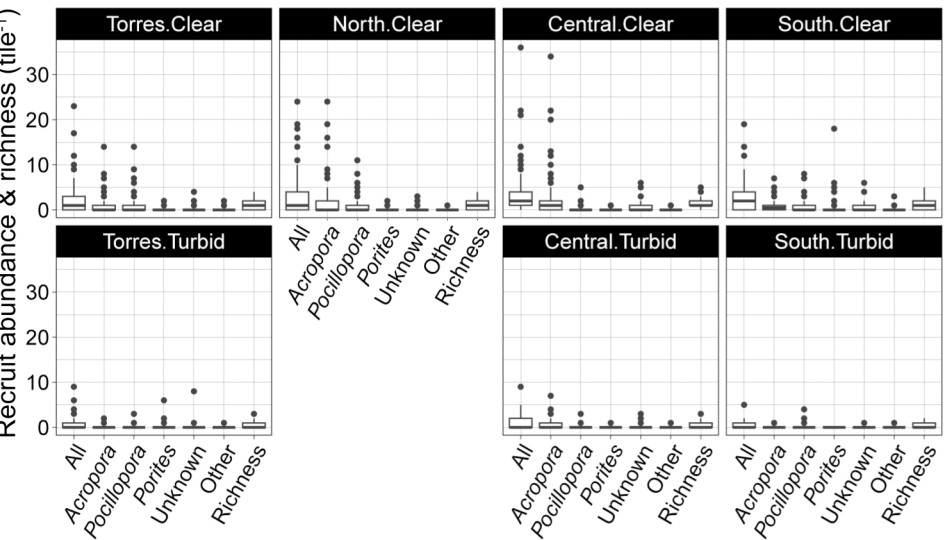

**Fig 4. Per tile coral recruit abundance, of common taxa, and recruit taxonomic richness, across the seven regions.** Values shown are the combined totals from all sides of the settlement tiles. One recruit per tile represents 75.6 recruits m$^{-2}$. Boxes = inner quartile range, central horizontal line = median, vertical lines = variability outside upper and lower quartiles, and points = outliers.

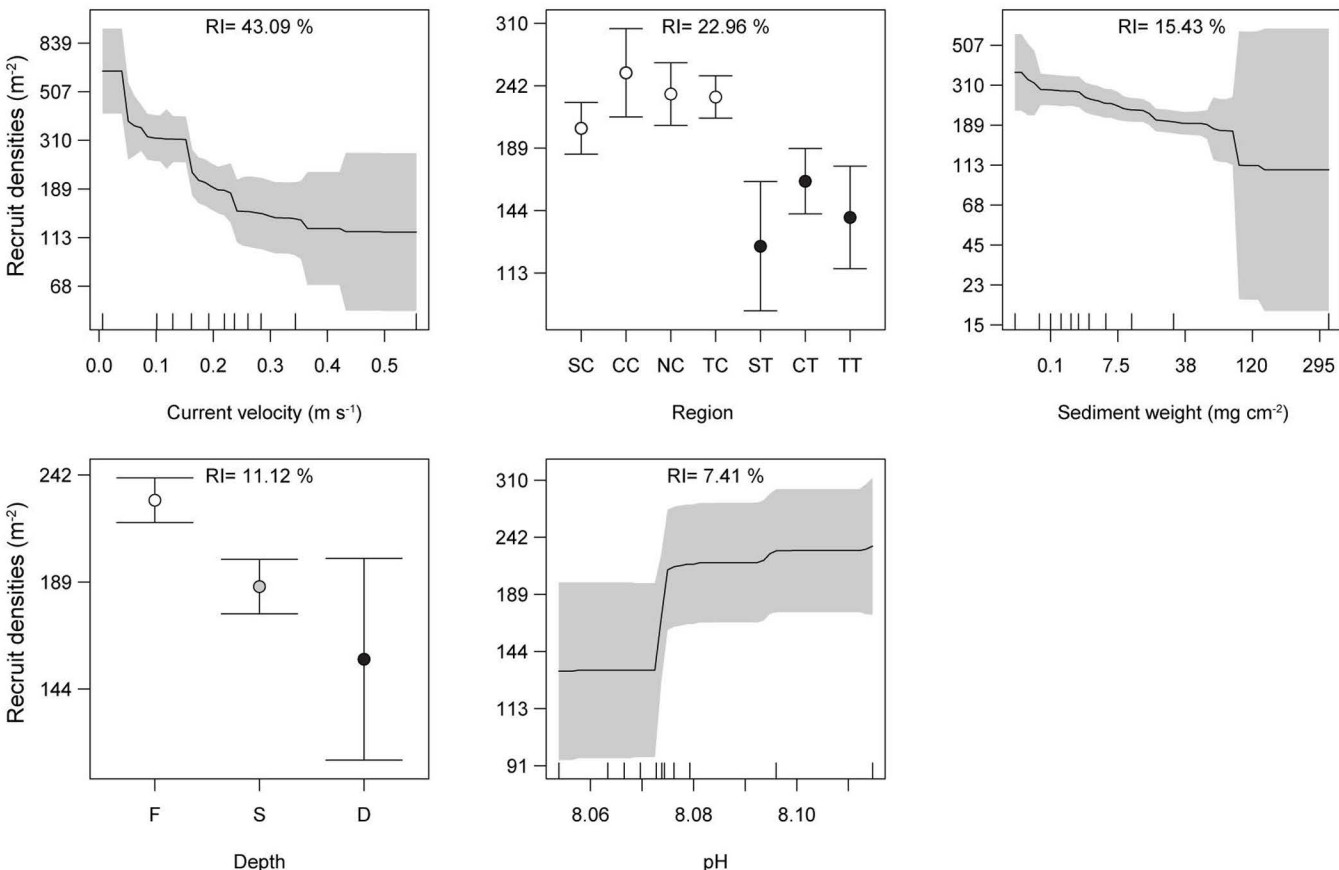

**Fig 5. Changes in coral recruit densities along long-term mean environmental conditions.** Partial dependence plots illustrate changes in recruit densities in relation to the strongest environmental parameters, and their relative influence (RI). The y-axis represents the change in recruit density attributable to each predictor. Solid lines and circles represent model means, grey envelopes and error bars are 95% confidence intervals. The ticks on the x-axis reflect the deciles of the predictors' distribution. Recruit densities are combined between the top, bottom, and sides of the tiles. Abbreviations: Regions: the first letter indicates latitude: South, Central, North and Torres Strait, the second letter indicates Clear or Turbid (see Fig 1). Depth: **F** = Reef flat (1 m), **S** = Shallow reef crest (5 m), **D** = Deep slope (15 m).

the deep slope (15 m depth) when all other parameters were held constant. When an ABT was performed for solely top of tile recruit density, this relationship flipped with top density being highest at the deep (15 m) slopes. Overall, densities were highest on reef flats in clear-water regions with low current velocity, low sedimentation, and high pH.

Recruit densities were also related to some of the benthic taxa on the tiles, but differences in densities were much smaller than those associated with the environmental variables (Fig 6). Turf cover on tops of tiles was the strongest benthic predictor (RI: 29.04%), where increasing turf cover was associated with increasing recruit density (up to ~ 130 recruits m⁻²). The three strongest benthic predictors (turf, CCA, and "bare" (PVC covered by biofilms but free of macrobenthos) tile) were all found on the tops of the tiles, despite recruits primarily being located on tile bottoms. Where top of tile CCA cover and bare tile space were low (>35% and > 10%, respectively), recruit densities were up to 50 recruits below average, and where CCA cover > 35% or bare tile space > 30%, recruit density was slightly above average. On the tile bottoms, intermediate fleshy invertebrate cover (<40%) was associated with slight (40 recruits m⁻²) increases in recruit density, and low *Peyssonnelia* (<15%) and CCA (<40%) cover were associated with

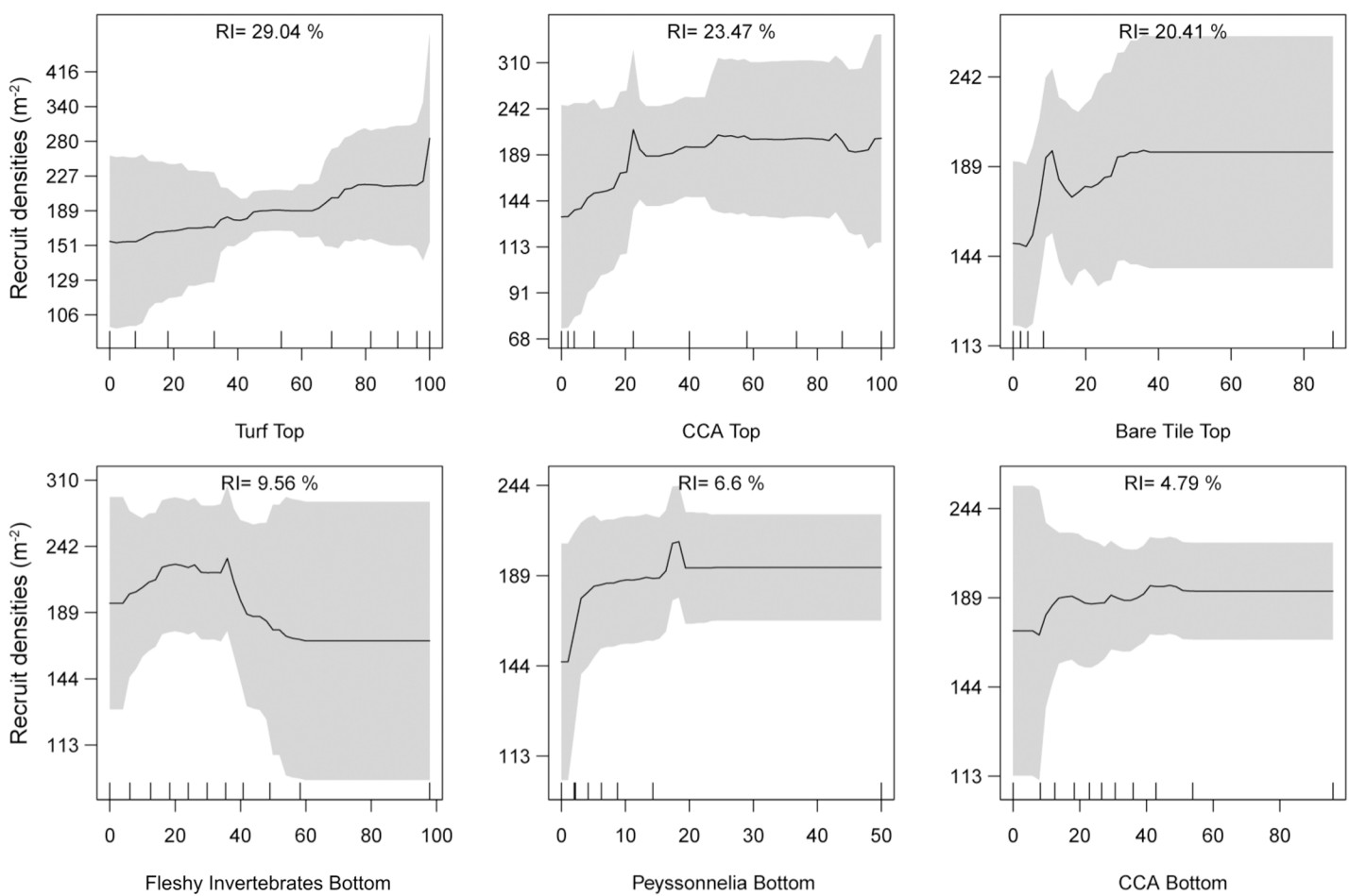

**Fig 6. Partial dependence plots of recruit densities associated with the cover of the main biota on the top and bottom of the tiles.** 'Top' and 'bottom' describe the tile position of the benthic taxa (legend as in Fig 5).

slightly lower than average recruit densities. Repeating the analyses just for the bottom recruits with bottom biota did not improve predictions and also led to inconclusive patterns.

An RDA (Fig 7) of the environmental and spatial predictors explained 26.20% of variance in tile benthos (including recruits) in its first two axes. It showed that high total recruit abundance was associated with reef flats, high aragonite saturation state, high total alkalinity, high Secchi depth, and the regions Central.Clear and Torres.Clear. Recruit abundances were also positively associated with the bottom cover of *Peyssonelia*, CCA and turfs; top CCA and top turf were orthogonal to recruit densities. Low recruit abundances were associated with environmental conditions of high total nitrogen, high current velocity, high sediment weight, and the turbid regions, alongside the cover of fleshy invertebrates, bottom bare tile, and top *Peysonnelia*. The negative associations between bare bottom tile and coral recruits were potentially due to bare bottom tiles being observed in turbid and sediment laden environments; in addition, 'bare tile' was occasionally an artifact when macro-benthos, e.g., large sponges adhered more to the reef than to the tile and were dislodged during tile collection. Bryozoa were found to be associated with high current velocity, salinity, and nutrient concentration.

The RDA suggested that the two most abundant top communities, CCA and turf, were in opposition to one another, reflecting competition for space. The variance of these community

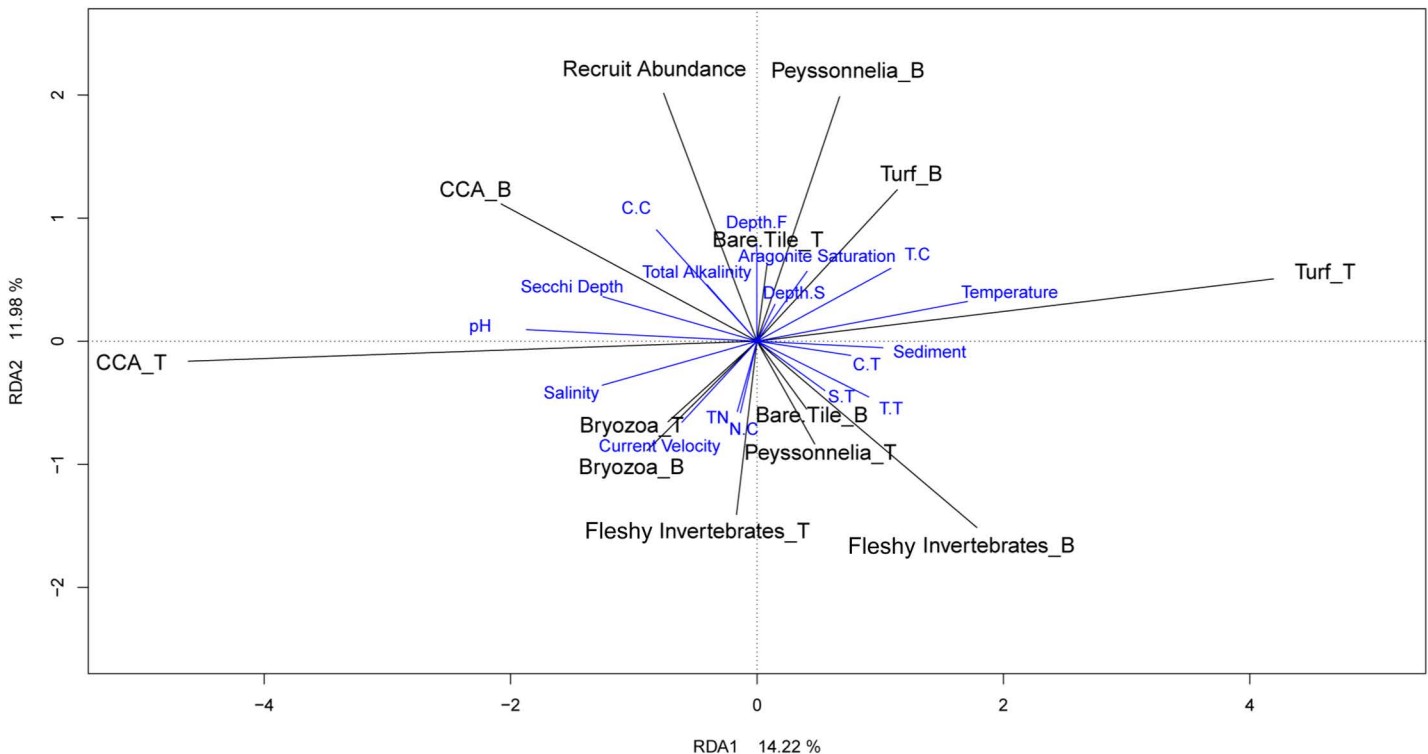

**Fig 7. Redundancy analysis (RDA) of the relationship between environmental and spatial predictors (blue vectors), the percent cover of benthic taxa and coral recruit abundances on the tiles.** Black vectors indicate the direction and magnitude of the benthic variables and recruits. Abbreviations: the suffix _T distinguishes taxa found on the top of tiles while _B indicates those found on the bottom of tiles; Regions and depths: see Fig 5; TN = total nitrogen.

components was primarily explained by RDA1, with top CCA being positively associated with higher pH, Secchi depth, salinity, current velocity (Ubed), and total alkalinity levels, and low sediment and temperature. Competition of space is also suggested on the tile bottom sides through the triangular distribution of bottom-tile CCA, turf, and fleshy invertebrate cover.

A permutation analysis confirmed the significant relationship of the benthic communities and recruit abundance to the environmental and spatial factors, with sediment accounting for the greatest F-ratio, followed by region, total nitrogen, depth, water velocity, pH, temperature, and aragonite saturation state (Table 1). Total alkalinity, Secchi depth, and salinity were deemed insignificant.

## Discussion

This study reveals key patterns in coral recruit densities, and identifies their main environmental, spatial, and benthic predictors for the Great Barrier Reef and Torres Straits. In addition, it identifies environmental conditions regulating patterns in key benthic taxa (CCA, turf, and fleshy invertebrates).

Across the wide range of GBR habitats investigated here, recruit densities averaged ~ 187 ± 12 recruits m$^{-2}$ (SE) (when disregarding 3D complexity). GBR recruit densities are generally higher than the global average (Table 2). Both on the GBR and globally, estimates of coral recruit densities derived from settlement tiles also vary upon tile submersion time (Table 2). Recruits on settlement tiles face rapid mortality within the first year (~80%) and declining settlement rates (reduction by ~ 50% per year), in part, attributable to overgrowth and loss of suitable settlement surfaces by successional benthic communities and changes in benthic

**Table 1. ANOVA like permutation of RDA results: benthic communities (functional groups and coral recruit abundance) modeled by environmental and spatial predictors.**

|  | DF | Variance | F | Pr(>F) |
|---|---|---|---|---|
| **Sediment Weight (g)** | 1 | 0.264 | 14.273 | 0.001 |
| **Region** | 6 | 1.313 | 11.849 | 0.001 |
| **TN** | 1 | 0.162 | 8.859 | 0.001 |
| **Depth** | 2 | 0.292 | 7.900 | 0.001 |
| **Current Velocity** | 1 | 0.129 | 6.973 | 0.001 |
| **pH** | 1 | 0.102 | 5.519 | 0.001 |
| **Temperature** | 1 | 0.092 | 4.998 | 0.001 |
| **Aragonite Saturation State** | 1 | 0.058 | 3.125 | 0.001 |
| **Total Alkalinity** | 1 | 0.034 | 1.829 | 0.055 |
| **Secchi Depth** | 1 | 0.019 | 1.043 | 0.374 |
| **Salinity** | 1 | 0.012 | 0.628 | 0.789 |
| **Residual** | 570 | 10.525 | – | – |

**Table 2. Coral recruit densities in the Great Barrier Reef (GBR) compared with all other regions (Global) on settlement tiles deployed for varying lengths of time, compared to the recruit density observed in this study. Shown are arithmetic means ± one standard error (in brackets: number of sites). Top rows are derived from the studies listed in Edmunds [73].**

| Deployment Duration (Months) | GBR coral recruits per m² | Global coral recruits per m² | Data Origin |
|---|---|---|---|
| <3 | 1810 ± 217 (87) | 1070 ± 209 (382) | Edmunds, 2023 |
| 3 to <6 | 1110 ± 178 (73) | 209 ± 57 (544) | Edmunds, 2023 |
| 6 to <12 | 666 ± 164 (43) | 90 ± 11 (292) | Edmunds, 2023 |
| 12 to 18 | 632 ± 209 (7) | 103 ± 10 (203) | Edmunds, 2023 |
| 24 | – | 44 ± 26 (113) | Edmunds, 2023 |
| 20 to 24 | 187 ± 12 (141) | – | This study |

composition [45]. These data suggest that while < 3 month old tiles are the most effective indicators of larval supply, the 24 months old settlement tiles are good proxies for coral recruitment as they closely resemble the surrounding reef benthos and hence integrate across larval supply, settlement and post-settlement survival [73].

Decreased coral recruitment was found in areas of high current velocity, which was found to be the most influential of all environmental predictors, displaying a reduction of 530 recruits m$^{-2}$ across the velocity gradient from 0.0062 to 0.57 m s$^{-1}$. Declining recruit densities with increasing water velocity and wave exposure has been previously noted for acroporid juveniles [4,74,75], which constituted the largest portion of recruits in the present study (51.91%). Reef habitat types were correlated to current velocity, with low current velocity lagoons observing the highest recruit densities, compared to high current velocity reef fronts. Reduced recruit density in high flow may be attributed to settlement bottlenecks, such as settlement inhibition and predation of larvae by planktivores, and passive larval transportation away from reefs in high-flow areas [41,76]. On the other hand, water velocity promotes coral growth, as it reduces the thickness of the diffusive boundary layer hence increasing fluxes of gas, nutrients, and metabolites between the coral and the surrounding water [77] and prevents the settlement of sediments. Hence despite its negative effect on recruit densities, water velocity can positively influence juvenile density; for instance, *Montipora* juveniles have been reported at higher densities in higher wave exposure, which was attributed to elevated CCA abundance [75]. Additionally, this study examined long-term mean current velocity at the

1-kilometer level; hydrodynamics at finer spatial scales including microhabitats and reef structure complexity likely play a further significant role in recruitment, as coral larvae are passive to currents and are influenced by eddies that facilitate settlement [76]. Our findings contribute to the limited studies targeting water velocity effects on recruit populations, highlighting the importance of flow conditions to recruitment along the GBR.

Water quality was another strong determinant of recruit densities, as indicated through multiple predictors. The classification of regions as either turbid or clear-water, the measured amount of sediment deposited on the tiles, and modelled long-term mean pH all showed a negative effect of poor water quality on coral recruitment. Recruit counts in the three turbid regions observed a median density of zero and were on average 3-fold lower compared to clear-water regions. Recruit densities were lowest in the South.Turbid region (Keppel Islands group), the second most turbid of the investigated regions (long-term mean Secchi depth across the study sites: 9.0 ± 0.37 m (SE), mean sediment dry weight: 21.7 ± 8.93 mg cm$^{-2}$ (SE)). Its reefs are located near the mouth of the major Fitzroy River, and, of the three turbid regions, are most influenced by human modification and the terrestrial runoff of sediments and nutrients. These reefs appear to be at a higher risk of slower recovery via recruitment than any of the other regions. The most turbid region, the Torres.Turbid (Dungeness Reef; 8.7 m ± 0.22 long-term mean Secchi depth, 33.3 ± 4.87 mg cm$^{-2}$ sediment weight), is located ~60 km off the coast of Papua New Guinea, and 130 km from the major Fly River. This region is little studied, but due to its remote location, its high turbidity is likely unrelated to human activity and instead possibly attributable to complex resuspending hydrodynamics in its vast shallow reef expanses. The Central.Turbid region (Palm Islands group) is located ~120 km downstream of the major Burdekin River and has the clearest water of the three turbid regions (long-term mean Secchi depth: 15.4 m ± 0.15, 28.9 ± 5.81 mg cm$^{-2}$ tile sediment weight), and the ABT showed the highest recruit numbers of the three regions once data were controlled for the other predictors. It remains to be investigated to what extent the low recruitment is a function of lower larval supply from self-seeding or upstream reefs, low larval settlement success, and low recruit survival in these three disparate turbid regions.

In addition to the regional differences, sedimentation was associated with strongly reduced coral recruit density. Fine sediments smother corals and coral recruits, reduce light availability and gas exchange, expose recruits to harmful bacteria, and can increase bleaching related recruit mortality rates [78–80]. Evidence suggests that even very thin (<150 μm) sediment layers alter settlement preferences and adversely affect coral recruitment success through settlement inhibition and smothering [14,29,81]. Indeed, we observed that recruit density declined monotonously with increasing sediments. For example, as sediment increased from 0.1 to 38 and 121 mg dry weight cm$^{-2}$, recruit densities observed a decline of ~130 and 245 recruits m$^{-2}$ respectively. The redundancy analysis suggested high turbidity (i.e., Secchi depth), from fine anthropogenically derived sediments, was negatively associated with recruit density (the environmental ABT also suggested an association, although slightly weaker than the other predictors). Declines in hard coral recruit and juveniles with turbidity has previously been noted [34,81–83]. Turbidity can significantly reduce fertilization, survival and growth of young corals symbiont productivity [14,84].

This study also corroborates existing literature on the reduction of coral recruit density with ocean acidity [33,34]. Below the pH of ~8.07, coral recruit density steeply declined. Similar negative relationships between pH and both recruit and juvenile density have been observed along CO$_2$ seeps [33,85] and in the GBR [34]. *Acropora*, *Pocillopora*, and even *Porites*, which made up the majority of recruits (91.55%), are known to be sensitive to ocean acidification, especially in the recruitment phase [33,85,86]. At more extreme levels, increased acidity is known to reduce and disrupt larval settlement and causes declines in calcification

rates and skeletal deformities [87,88]. Here, coral recruit density was sensitive to small changes in long-term mean pH levels, with densities reduced by approximately ~ 100 recruits m$^{-2}$ between both ends of the pH gradient. At present pH levels, pH had the lowest relative influence (RI: 7.41%) of the final short list of long-term environmental parameters, indicating that regional water quality differences and local sedimentation rates are more important drivers of recruit density, although pH is declining rapidly as atmospheric $CO_2$ continues to increase.

At fine spatial scales, our study confirms the influence of depth on recruit density. Total recruit density, of which 83.8% are located on tile undersides, declined with increasing depth; meanwhile, top of tile coral recruit density increased with depth. Corals tend to settle on tile undersides in shallow, tropical waters in part to avoid grazing and sedimentation, but choices shift to upper surfaces where light is limiting [12,13]. Indeed, research suggests light as the principle factor influencing orientation choices [89,90]. The exclusion of light as a predictor in our study was due to its confounded nature with region, depth and turbidity. In comparison to deep and shallow tiles, recruit density was greatest at the reef flats (1 m depth), despite this depth observing the highest water velocity. These results may indicate that bottom sides of tiles may act as a refuge to high water velocity at shallow depths.

For the benthic taxa, the ABT suggested an overall relatively minor effect on recruit densities, each accounting for differences by only 30 to 120 recruits per m$^2$. They suggested that the top communities of turf, CCA, and bare tile had a slightly stronger influence on recruit density than other benthic taxa, despite only 6.1% of recruits being found on the tops of tiles. Increasing top turf cover was associated with increasing recruit numbers, and < 20% top CCA was associated with lower recruit numbers. In contrast, the RDA indicated a positive relationship of recruit abundances to bottom *Peyssonelia* and CCA, a negative relationship to bottom and top fleshy invertebrates, and an orthogonal relationship to top CCA and top turf. Both analyses suggested that environmental conditions had a greater impact on total recruit density than community coverages. We hypothesize that top communities act as proxies for general environmental and water quality conditions that are important in defining total recruit densities, thus making them indirect benthic predictors of recruit density. This is due to communities on the tops of tiles being more exposed to environmental conditions, especially sedimentation and current velocity, compared to the less exposed bottom communities.

CCA cover was positively associated with the same water quality conditions that favored coral recruits (low sediment, high Secchi depth, and high pH). An identification of CCA to species or genus level would have likely further strengthened the predictive power of CCA [28] but was beyond the scope of this study. Generally, CCA cover has been found to increase away from the coast in line with improved water quality conditions [34,91]. High sediment and turbidity may disrupt CCA settlement, reduce light availability, and increase the presence of competitors such as turf algae, contributing to the decline in CCA cover under poor water quality [92–95]. CCA is known to significantly decline in reduced pH and aragonite environments [34,96]. Our results indicate that CCA is an important water quality indicator, and associated declines in coral recruits with diminished CCA cover suggests that CCA predicts areas of poor water quality sensitive to coral recruits. In addition, as a preferred settlement substrate and inhibitor of coral competitors [28,43,44], elevated CCA cover may contribute to recruit densities.

Interactions among algal turfs and coral recruits are complex and not entirely understood; here, we modeled positive associations between turf algal cover and coral recruit density. These results must be taken with caution since percent cover is a 2D metric that does not capture important 3D features of turf. Factors contributing to the direction of interaction include density, height, presence of sediments, and herbivory strength [39,97,98]. For instance, the combined presence of long turfs and sediments has a strong negative effect on coral settlement

and can result in near absence of recruits [39,97]. On the other hand, short sparse turfs can reduce the risk of parrotfish grazing to coral juveniles, although these effects may be outweighed by algal-coral competition in the long term [98]. In addition, sparse turf is comparatively less detrimental to recruit survival than *Peyssonnelia* spp. overgrowth [37]. Our results of a positive association between recruit abundances and turfs does not elicit information on potential direct or indirect mechanistic links. Some types of turfs found on the GBR may be less inhibitive than some other forms of benthos, which may outweigh the turfs' sediment trapping properties. Also, algal turfs are generally ephemeral, and their presence may have been coincidentally higher at the time of observation compared to the time of coral settlement, artificially creating a positive association between turf and coral recruits.

Fleshy invertebrates (sponges and ascidians) were the most influential benthic taxa on the tile bottoms. Coral recruit density was greatest when fleshy invertebrate cover was lower than 40%, where density fell below average by roughly 20 recruits m$^{-2}$ suggesting some competitive exclusion. Indeed, Arnold and Steneck [45] observed declining recruitment rates in the Caribbean with increasing invertebrate (sponge, ascidian, byrozoan) dominance on settlement tiles over time. Bottom fleshy invertebrates, which are all filter feeders, responded positively to poor water quality conditions (high sediment, low Secchi depth, and low pH), where recruit densities were reduced; this pattern, alongside that of declining CCA, again emphasizes the pervasive direct and indirect effects of water quality, profoundly altering ecological balances in coral reefs.

Examining recruit and juvenile densities is important in the context of reef recovery, which relies on the influx of new individuals to the population and regrowth of existing colonies. Through an 11-year time series across GBR inshore reefs, Thompson et al. [99] found that beyond the threshold of 4.6 juveniles per m$^2$ (juveniles defined as < 5 cm) there was a > 50% probability of recovery of coral cover, and at 13 juveniles per m$^2$, this probability increased to > 80% [99]. Graham et al. [100] found similar results, where densities > 6.2 per m$^2$ juveniles (<10 cm diameter) reduced the probability of a regime shift in the Seychelles. The average recruit densities on the settlement tiles reported here (region-wide averages 43.5 ± 12 (SE) to 247 ± 32 recruits m$^{-2}$) were one to two orders of magnitude greater than these suggested thresholds for coral juveniles. Comparisons between coral recruit and juvenile densities strongly depend on their survival to the juvenile stage, as well as year-to-year variation in recruitment. In the first year recruit mortality ranges from > 30 to 99%, although mortality rates tend to decline once recruits reach a size escape threshold (generally 5 mm diameter or 3-9 months old) (as reviewed in Randall et al. [101]). Indeed, some studies found densities of recruits on settlement tiles not to be good indicators of reef recovery [83]; our findings suggest that tiles deployed for 2 years do not show such limitations. Our tiles provided useful information on differences in relative coral recruit densities, indicating areas where recovery may be facilitated or limited by recruitment. Our data show that recruits would require a 34% survival rate in the South.Turbid (lowest recruit density) yet only a 5% survival in the Central.Clear (greatest density) to achieve 80% probability of recovery, according to Thomson et al. [99] thresholds. Notably, all turbid regions had a median of zero recruits on the tiles, emphasizing the elevated risk of slow recovery in turbid environments.

Our study was the first to investigate settlement tile communities with the AI platform ReefCloud, which has been optimized to auto-annotate coral reef communities. The use of AI in benthic-point classification was indispensable to deal with the large number of tiles, as it vastly decreased data processing times, a major bottleneck in marine ecological sampling [102–104]. This novel use of AI may, however, pose as a potential limitation to the present study. While the most influential benthic groups to coral recruitment (CCA and turf algae) were recognized with high accuracy (82 and 87%), confusion between some common taxa

occurred (e.g., sponges and ascidians). Grouping their many species under one label increases variations in morphologies (such as color, texture, size) that define the taxa, adding confusion to the model (as observed in González-Rivero et al. [58]). Combining data into functional groups such as fleshy invertebrates representing both the sponges and ascidians, and the use of human identification (approximately 18% of the data set) reduced confusion in our resulting data, with a total estimated weighted average accuracy of 80%.

In conclusion, this study stresses the importance of current velocity and water quality to coral recruitment success, outlining areas of high coral recruit densities and areas that are relatively more recruitment limited. Current velocity described the largest variation in recruit density across the seven regions of the GBR and Torres Strait. While we found that coral recruit densities on settlement tiles were likely above those previously defined thresholds for putatively positive/expected reef recovery rates at all reefs, those thresholds had been defined based on densities of juvenile corals on the reef benthos, and location and taxon-specific survival probabilities are still poorly understood. Densities in turbid waters were on average three times lower than in clear-water regions despite the diverse nature of turbid regions represented in this study. Low water quality reefs are more susceptible to disease and crown of thorns outbreaks, with bleaching projections revealing that water quality improvements will be necessary to maintain future coral resilience [105]. Where natural recovery speed becomes insufficient, human intervention through reef restoration is now being considered in the GBR [106], where it will have to target the areas of greatest need and the likelihood of greatest ecological benefits. Our findings suggest that these are especially deeper reef sites in turbid regions where sediment deposits are high. High-current areas will need additional assessments as they tend to be associated with high growth, survival and resilience, potentially compensating for their demonstrated low recruit densities. Our data informs restoration practitioners on drivers of natural recruit densities, as baseline against which to plan coral deployment densities, once more information on early recruit and juvenile survival are known. Our data suggest that turbid reefs, areas of high sedimentation, high nutrients, and low pH may be the first to ecologically benefit from active management. However, it first and foremost remains imperative to reduce climate and water quality threats alongside restoration, especially as the severity of these threats will only increase in the coming decades [2].

## Supporting information

**S1 Fig. Tiles *in situ*, after two years of deployment, from clear regions, taking on characteristic benthos of surrounding reef substrata.**
(PDF)

**S2 Fig. Tiles *in situ*, after two years of deployment, from turbid regions, taking on characteristic benthos of surrounding reef substrata.**
(PDF)

**S1 Table. Number of settlement tiles retrieved across the 7 regions, and 12 reefs, per site and depth.** Acronyms: FR = front reef, FL = flank reef, BA = back reef, LA = lagoon, D = deep (15m), S = shallow (5m), F = flat (1m).
(DOCX)

**S2 Table. Environmental and spatial predictors.**
(DOCX)

**S3 Table. Label set categories used in ReefCloud to annotate the settlement tile communities, and their categorization into benthic functional groups used in the study.**
(DOCX)

**S3 Fig. Confusion matrix portraying points classified by ReefCloud (Y) as a percentage of matching human classifications (X), using the original label set (Table 1).** Human classified points were derived from a randomized subset of 25% of tiles. Numbers on the x-axis label indicate the number of points visually annotated per class. Values along the diagonal indicate the percentage of points accurately labeled by ReefCloud, values that vertically deviate from the diagonal line indicate a misclassification by ReefCloud. Cells with values ≥1% are labeled.
(PDF)

**S4 Fig. Categorization of visually annotated ('actual') benthic functional groups (Table 1) in comparison to machine predicted categorizations.** Mean weighted accuracy = 76.1%. (Legend as in S3 Fig).
(PDF)

## Acknowledgements

We acknowledge the Aboriginal and Torres Strait Islander Traditional Owners of the sea Country where this work was conducted and are grateful for their consent to conduct this research. We thank Christopher Doropoulos and Renata Ferrari for their invaluable contributions in reviewing an earlier version of this manuscript, and to Mathilda Bates for her assistance in data collection.

## Author contributions

**Conceptualization:** Katharina E. Fabricius.

**Data curation:** Matilde Drake, Sam H. C. Noonan, Mariana Alvarez-Noriega.

**Formal analysis:** Matilde Drake, Mariana Alvarez-Noriega, Ahmad R. Rashid, Katharina E. Fabricius.

**Funding acquisition:** Katharina E. Fabricius.

**Investigation:** Matilde Drake.

**Methodology:** Matilde Drake, Sam H. C. Noonan, Mariana Alvarez-Noriega, Katharina E. Fabricius.

**Validation:** Matilde Drake.

**Visualization:** Matilde Drake, Sam H. C. Noonan, Ahmad R. Rashid, Katharina E. Fabricius.

**Writing – original draft:** Matilde Drake, Katharina E. Fabricius.

**Writing – review & editing:** Matilde Drake, Sam H. C. Noonan, Mariana Alvarez-Noriega, Ahmad R. Rashid, Katharina E. Fabricius.

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
