## [Decision Letter · Decision Letter 0]

29 Oct 2024

PONE-D-24-35629Current velocity, water quality, and benthic taxa as predictors for coral recruitmentPLOS ONE

Dear Dr. Drake,

Thank you for submitting your manuscript to PLOS ONE. After careful consideration, we feel that it has merit but does not fully meet PLOS ONE’s publication criteria as it currently stands. Therefore, we invite you to submit a revised version of the manuscript that addresses the points raised during the review process. Please submit your revised manuscript by Dec 13 2024 11:59PM. If you will need more time than this to complete your revisions, please reply to this message or contact the journal office at plosone@plos.org . Please include the following items when submitting your revised manuscript:

We look forward to receiving your revised manuscript.

Kind regards,

Tzen-Yuh Chiang

Academic Editor

PLOS ONE

Journal Requirements: When submitting your revision, we need you to address these additional requirements. 1. Please ensure that your manuscript meets PLOS ONE's style requirements, including those for file naming. The PLOS ONE style templates can be found at https://journals.plos.org/plosone/s/file?id=wjVg/PLOSOne_formatting_sample_main_body.pdf and https://journals.plos.org/plosone/s/file?id=ba62/PLOSOne_formatting_sample_title_authors_affiliations.pdf 2. We note that the grant information you provided in the ‘Funding Information’ and ‘Financial Disclosure’ sections do not match.  When you resubmit, please ensure that you provide the correct grant numbers for the awards you received for your study in the ‘Funding Information’ section. 3. Thank you for stating the following in the Acknowledgments Section of your manuscript: "This research was supported by the Reef Restoration and Adaptation Program, a partnership  between the Australian Governments Reef Trust and the Great Barrier Reef Foundation. We acknowledge the Aboriginal and Torres Strait Islander Traditional Owners of the sea Country  where this work was conducted and are grateful for their consent to conduct this research. We would like to thank Christopher Doropoulos and Renata Ferrari for their invaluable contributions in reviewing an earlier version of this manuscript, and to Mathilda Bates for her assistance in data collection." We note that you have provided funding information that is not currently declared in your Funding Statement. However, funding information should not appear in the Acknowledgments section or other areas of your manuscript. We will only publish funding information present in the Funding Statement section of the online submission form. Please remove any funding-related text from the manuscript and let us know how you would like to update your Funding Statement. Currently, your Funding Statement reads as follows: "This research was funded by the Reef Restoration and Adaptation Program, a partnership between the Australian Governments Reef Trust and the Great Barrier Reef Foundation (https://gbrrestoration.org/), and the Australian Institute of Marine Science (https://www.aims.gov.au/). The funders had no role in study design, data collection and analysis, decision to publish, or preparation of the manuscript." Please include your amended statements within your cover letter; we will change the online submission form on your behalf. 4. When completing the data availability statement of the submission form, you indicated that you will make your data available on acceptance. We strongly recommend all authors decide on a data sharing plan before acceptance, as the process can be lengthy and hold up publication timelines. Please note that, though access restrictions are acceptable now, your entire data will need to be made freely accessible if your manuscript is accepted for publication. This policy applies to all data except where public deposition would breach compliance with the protocol approved by your research ethics board. If you are unable to adhere to our open data policy, please kindly revise your statement to explain your reasoning and we will seek the editor's input on an exemption. Please be assured that, once you have provided your new statement, the assessment of your exemption will not hold up the peer review process. 5. Please include your full ethics statement in the ‘Methods’ section of your manuscript file. In your statement, please include the full name of the IRB or ethics committee who approved or waived your study, as well as whether or not you obtained informed written or verbal consent. If consent was waived for your study, please include this information in your statement as well. 6. We note that Figure 1 in your submission contain [map/satellite] images which may be copyrighted. All PLOS content is published under the Creative Commons Attribution License (CC BY 4.0), which means that the manuscript, images, and Supporting Information files will be freely available online, and any third party is permitted to access, download, copy, distribute, and use these materials in any way, even commercially, with proper attribution. For these reasons, we cannot publish previously copyrighted maps or satellite images created using proprietary data, such as Google software (Google Maps, Street View, and Earth). For more information, see our copyright guidelines: http://journals.plos.org/plosone/s/licenses-and-copyright. We require you to either (1) present written permission from the copyright holder to publish these figures specifically under the CC BY 4.0 license, or (2) remove the figures from your submission: A. You may seek permission from the original copyright holder of Figure 1 to publish the content specifically under the CC BY 4.0 license.   We recommend that you contact the original copyright holder with the Content Permission Form (http://journals.plos.org/plosone/s/file?id=7c09/content-permission-form.pdf) and the following text:“I request permission for the open-access journal PLOS ONE to publish XXX under the Creative Commons Attribution License (CCAL) CC BY 4.0 (http://creativecommons.org/licenses/by/4.0/). Please be aware that this license allows unrestricted use and distribution, even commercially, by third parties. Please reply and provide explicit written permission to publish XXX under a CC BY license and complete the attached form.” Please upload the completed Content Permission Form or other proof of granted permissions as an ""Other"" file with your submission. In the figure caption of the copyrighted figure, please include the following text: “Reprinted from [ref] under a CC BY license, with permission from [name of publisher], original copyright [original copyright year].” B. If you are unable to obtain permission from the original copyright holder to publish these figures under the CC BY 4.0 license or if the copyright holder’s requirements are incompatible with the CC BY 4.0 license, please either i) remove the figure or ii) supply a replacement figure that complies with the CC BY 4.0 license. Please check copyright information on all replacement figures and update the figure caption with source information. If applicable, please specify in the figure caption text when a figure is similar but not identical to the original image and is therefore for illustrative purposes only.The following resources for replacing copyrighted map figures may be helpful: USGS National Map Viewer (public domain): http://viewer.nationalmap.gov/viewer/The Gateway to Astronaut Photography of Earth (public domain): http://eol.jsc.nasa.gov/sseop/clickmap/Maps at the CIA (public domain): https://www.cia.gov/library/publications/the-world-factbook/index.html and https://www.cia.gov/library/publications/cia-maps-publications/index.htmlNASA Earth Observatory (public domain): http://earthobservatory.nasa.gov/Landsat: http://landsat.visibleearth.nasa.gov/USGS EROS (Earth Resources Observatory and Science (EROS) Center) (public domain): http://eros.usgs.gov/#Natural Earth (public domain): http://www.naturalearthdata.com/ 7. Please review your reference list to ensure that it is complete and correct. If you have cited papers that have been retracted, please include the rationale for doing so in the manuscript text, or remove these references and replace them with relevant current references. Any changes to the reference list should be mentioned in the rebuttal letter that accompanies your revised manuscript. If you need to cite a retracted article, indicate the article’s retracted status in the References list and also include a citation and full reference for the retraction notice.

Reviewers' comments:

Reviewer's Responses to Questions

**Comments to the Author**

1. Is the manuscript technically sound, and do the data support the conclusions?

Reviewer #1: Yes

2. Has the statistical analysis been performed appropriately and rigorously? 

Reviewer #1: Yes

3. Have the authors made all data underlying the findings in their manuscript fully available?

Reviewer #1: Yes

4. Is the manuscript presented in an intelligible fashion and written in standard English?

Reviewer #1: Yes

5. Review Comments to the Author

Reviewer #1: This manuscript examined the effect of environmental factors on the recruitment of corals in Great Barrier Reef. The study is well designed and is the first to use AI to score the faunal density on the settlement tiles. I think the MS can be accepted after minor revision as below:

1) On the use of the settlement plates, the authors claimed that the use of PVC materials can result in settlement pattern more similar to natural assemblages. However, Nozawa et al. (2011) found that plates with crevices can also enhance coral recruits. Although this study has not used this kind of coral recruit settlement plate, I would suggest the author to cite this reference that plates with crevice microstructures can enhance coral recruitment.

Nozawa et al 2011 - Reconsideration of the Surface Structure of Settlement Plates Used in Coral Recruitment Studies – Zoological Studies 50(1): 53-60

2) Any morphological features to aid identification of coral recruits? Have this features used in training the AI? I would suggest you can cite the reference for recruit identifications. An example is:

Babcock et al. 2003 - Identification of Scleractinian Coral Recruits from Indo-Pacific Reefs. Zoological Studies 42(1): 211-226

3) The authors pointed out effect of environmental factors on coral recruitment received scant attentions. I would suggest in addition to the statistical analysis conducted in the present study, to use multivariate analysis to generate a separated nMDS patterns for the environmental and biological assemblages. This can be use the multivariate software PRIMER. Then use the Bio-Env analysis to match the environmental and biological assemblages to see any significant correlation and deduce the best subset of env. Factors which are more responsible for the biological assemblages. This result can further strengthen the PCA plots which the author conducted in the current manuscript. The strong side of this analysis can reflect whether the environment and biological assemblages are different among the sites and then correlate the two patterns together.

4) Effect of light on coral recruitments, there are examples in the Pacific that light can affect recruitment as the tropical and subtropical latitudinal waters differs in light intensity. The authors can consider reading and citing the following relevant references in their discussion for comparisons.

Maida MJ, Coll C, Sammarco PW: Shedding light on scleractinian coral recruitment. J Exp Mar Biol Ecol 1994, 180: 189–202. 10.1016/0022-0981(94)90066-3

Maida M, Sammarco PW, Coll JC: Effects of soft corals on scleractinian coral recruitment. I: directional allelopathy and inhibition of settlement. Mar Ecol Prog Ser 1995, 121: 191–202.

Ho, MJ., Dai, CF. Coral recruitment of a subtropical coral community at Yenliao Bay, northern Taiwan. Zool. Stud. 53, 5 (2014). https://doi.org/10.1186/1810-522X-53-5

5) What is the unit in the Y axis of figure 4?

6. PLOS authors have the option to publish the peer review history of their article (what does this mean? ). If published, this will include your full peer review and any attached files.

**Do you want your identity to be public for this peer review?** For information about this choice, including consent withdrawal, please see our Privacy Policy .

Reviewer #1: No

---

## [Author Response · Author response to Decision Letter 1]

13 Jan 2025

Dear editors Tzen-Yuh Chiang and Ronalyn M. Ramos and anonymous reviewer,

Thank you for taking the time to review our manuscript “Current velocity, water quality, and benthic taxa as predictors for coral recruitment.” We are grateful to have received your constructive comments, which have helped us improve our manuscript. We have thoroughly reviewed our manuscript and taken into consideration each of your comments. Below, we have responded to each comment individually with our response in italics and examples of changes to the text in blue.

In addition to changes requested by the editor and reviewer, we have also made minor changes to further improve clarity in the manuscript. We have changed the title from “Current velocity, water quality, and benthic taxa as predictors for coral recruitment” to “Current velocity, water quality, and benthic taxa as predictors for coral recruitment rates on the Great Barrier Reef”, better placing our study. Minor changes in word choice have been made throughout the text for greater clarity, for example we changed the name of one of our benthic predictors from “invertebrates” to “fleshy invertebrates” to emphasize the separation between this functional group (ascidians and sponges) to that of bryozoans. In addition, we have updated Table 2 recruit densities from global (global regions including GBR) to global (all other regions, excluding the GBR) to emphasize the differences in recruit density between the GBR and other global regions.

Once again, thank you for your valuable input. We look forward to your comments on the revised version of our submission.

With regards to PLOS ONE journal requirements:

Response: We have thoroughly reviewed PLOS ONE’s style requirements and have made minor changes to adhere to these standards. All instances of “Figure” have been changed to “Fig”. Supporting information captions have been moved below the References section. Text in Fig1 was not an approved font and has promptly been changed to the approved Arial font. We believe that the manuscript, including file names, now follows all PLOS ONE’s style requirements.

Response: The Financial Disclosure is correct and the Funding Information section of the application will be edited upon revised manuscript submission.

3. We note that you have provided funding information [in the Acknowledgments Section] that is not currently declared in your Funding Statement. However, funding information should not appear in the Acknowledgments section or other areas of your manuscript. We will only publish funding information present in the Funding Statement section of the online submission form.

Please remove any funding-related text from the manuscript and let us know how you would like to update your Funding Statement.

Response: The funding information we have provided in the Funding Statement is correct, and we have now deleted the funding statement from the Acknowledgements: “We acknowledge the Aboriginal and Torres Strait Islander Traditional Owners of the sea Country where this work was conducted and are grateful for their consent to conduct this research. We thank Christopher Doropoulos and Renata Ferrari for their invaluable contributions in reviewing an earlier version of this manuscript, and to Mathilda Bates for her assistance in data collection.”

Response: Yes, we understand that allowing access to data after acceptance may hold up publication timelines. We will be allowing public availability of our data sets which are prepared and ready to be published for a speedy upload upon acceptance.

Response: We have added “Ethics statement” to the “Methods” section of the manuscript. This section adds the permit issued by the Great Barrier Reef Marine Park Authority (G21/44774.1) to conduct our field research and sample collections. We have highlighted that our study did not require an animal ethics committee approval as targeted and sampled organisms were non-cephalopod and non-crustacean invertebrates and are not under the definition of “animal” (i.e. non-human vertebrate or cephalopod).

6. We note that Figure 1 in your submission contain [map/satellite] images which may be copyrighted. All PLOS content is published under the Creative Commons Attribution License (CC BY 4.0), which means that the manuscript, images, and Supporting Information files will be freely available online, and any third party is permitted to access, download, copy, distribute, and use these materials in any way, even commercially, with proper attribution. For these reasons, we cannot publish previously copyrighted maps or satellite images created using proprietary data, such as Google software (Google Maps, Street View, and Earth). AND Please clarify which map types you have used as https://s2maps.eu/# lists some as CC-BY SA or CC-BY NC which is not compliant with our licensing rules.

Response: We apologize for the potential misunderstanding of the licensure availability of our sources in Fig 1. Data sources used from EOX IT Services GmbH (Sentinel-2 Cloudless 2016) and AIMS, per these sources EOX IT (https://s2maps.eu), AIMS Sentinel 2 Composite (https://eatlas.org.au/geonetwork/srv/eng/catalog.search#/metadata/c38d2227-25c0-4d1e-adbc-bddb4aac1929/formatters/xsl-view?root=div&view=advanced), are CC BY 4.0 and are the only sources used in the updated Figure 1. Fig 1 has been simplified with the removal of unmodified 2024 Copernicus Sentinel-2 data. Fig 1 caption has been reworded to emphasize that all data used are under CC BY 4.0 license and clarifies which EOX IT cloudless layer was used (2016).

Response: We do not need to cite retracted articles but have added several new references in response to the reviewer and when doing a final reference completion check.

With regards to the anonymous reviewers’ main feedback:

1. On the use of the settlement plates, the authors claimed that the use of PVC materials can result in settlement pattern more similar to natural assemblages. However, Nozawa et al. (2011) found that plates with crevices can also enhance coral recruits. Although this study has not used this kind of coral recruit settlement plate, I would suggest the author to cite this reference that plates with crevice microstructures can enhance coral recruitment.

Nozawa et al 2011 - Reconsideration of the Surface Structure of Settlement Plates Used in Coral Recruitment Studies – Zoological Studies 50(1): 53-60

Response: Micro-crevices assist with mimicking natural rugosity in short term tile deployment studies where benthic communities are not well developed on the tiles, helping to enhance coral recruitment. But recent research (dela Cruz and Harrison, 2024; doi:10.1016/j.jembe.2024.152029) has shown that over time, "flat" surfaces like PVC get bio fouled which creates rugosity and makes differences among different tile materials in long term studies such as ours negligible.

2. Any morphological features to aid identification of coral recruits? Have this features used in training the AI? I would suggest you can cite the reference for recruit identifications. An example is:

Babcock et al. 2003 - Identification of Scleractinian Coral Recruits from Indo-Pacific Reefs. Zoological Studies 42(1): 211-226

Response: Babcock et al. 2003 is indeed the source our coral ID observers used to aid in morphological identification of the coral recruits. We have added this reference to the methods section. The AI was not able to perform at the genera level of taxonomic resolution, although it could identify “large” areas of recruits as hard or soft coral. This was due to the fact that many recruits were on the millimeter scale (undetectable in our images) and required microscopy to be identified.

3. The authors pointed out effect of environmental factors on coral recruitment received scant attentions. I would suggest in addition to the statistical analysis conducted in the present study, to use multivariate analysis to generate a separated nMDS patterns for the environmental and biological assemblages. This can be use the multivariate software PRIMER. Then use the Bio-Env analysis to match the environmental and biological assemblages to see any significant correlation and deduce the best subset of env. Factors which are more responsible for the biological assemblages. This result can further strengthen the PCA plots which the author conducted in the current manuscript. The strong side of this analysis can reflect whether the environment and biological assemblages are different among the sites and then correlate the two patterns together

Response: We appreciate your recommendation to use separate nMDS analyses and the Bio-Env approach to explore the relationship between environmental and biological assemblages.

Here, we have used boosted regression trees (BRT), which automatically select the most important environmental predictors based on their ability to explain variation in biological data. This method is particularly advantageous as it can account for complex, non-linear relationships and interactions between environmental variables and biological assemblages, making it a more robust approach for identifying key environmental drivers compared to traditional nMDS analysis.

Additionally, we employed Redundancy Analysis (RDA), which visually represents the relationship between environmental factors and biological assemblages in a 2D space. RDA provides a clear summary of how environmental variables influence biological patterns. To further substantiate these findings, we conducted PERMANOVA to test the statistical significance of the relationship between the environmental and biological data, thus confirming the strength of the patterns observed.

We believe that the combination of BRT, RDA, and PERMANOVA sufficiently addresses the reviewer's request by identifying important environmental predictors and demonstrating their relationship with biological assemblages. These methods allow us to provide a comprehensive understanding of the environmental influences on coral recruitment, without the need for separate nMDS or Bio-Env analyses.

4. Effect of light on coral recruitments, there are examples in the Pacific that light can affect recruitment as the tropical and subtropical latitudinal waters differs in light intensity. The authors can consider reading and citing the following relevant references in their discussion for comparisons.

Maida MJ, Coll C, Sammarco PW: Shedding light on scleractinian coral recruitment. J Exp Mar Biol Ecol 1994, 180: 189–202. 10.1016/0022-0981(94)90066-3

Maida M, Sammarco PW, Coll JC: Effects of soft corals on scleractinian coral recruitment. I: directional allelopathy and inhibition of settlement. Mar Ecol Prog Ser 1995, 121: 191–202.

Ho, MJ., Dai, CF. Coral recruitment of a subtropical coral community at Yenliao Bay, northern Taiwan. Zool. Stud. 53, 5 (2014). https://doi.org/10.1186/1810-522X-53-5

Response: Thank you for this suggestion, two of these references have now been added to the manuscript.

5. What is the unit in the Y axis of figure 4?

Response: The y-axis of Fig 4 reflects the total recruit abundance or richness (#) per tile. The y-axis has been changed to “Recruit abundance and richness (tile-1)” for clarity.

With regards to the reviewers’ attached comments (by line of the first submission):

1. Line 41: Include their ecological significance and cultural.

Response: Done, we have now included more information on coral reefs ecological and cultural significance as follows: “Coral reefs are the Earth’s most diverse marine ecosystems, serving as a critical habitat and nursery to thousands of species while providing socioeconomic benefits to millions of people through for instance, cultural heritage, fisheries, tourism, protection against coastal erosion [1]”

2. Line 43: Explain what types of disturbances.

Response: Thank you for this suggestion, we have now added more specifics of types of disturbances affecting coral reefs as follows: “…disturbances, such as cyclones, marine heat waves, mass bleaching, disease, and crown of thorns outbreaks, quashing coral populations”

3. Line 49: I think the intro paragraph needs to be more captivating

Response: We have now substantially rewritten the Introduction section, and believe it is now more captivating and outlining a better rationale for our study.

4. Line 55: This paragraph is quite short. Either explain more with details or include it with another paragraph

Response: Yes, we agree. We have now added more details to this paragraph on hydrodynamics (in the Introduction section).

5. Line 58: Sedimentation. Where is this sediment coming from? Erosion?

Response: While sources of sediment runoff can be tricky to pinpoint, research points towards erosion from land use change (particularly towards grazelands) to be the greatest contributor of increased of sediment loads. This information has now been added to the text. Changed word choice of sediments to sedimentation.

6. Line 59: Reefs closer to coastal zones may be more affected by sedimentation and nutrient run-off. Include a reference.

Response: Agreed. This section of the introduction was altered and more details and references on the origins of sediment which may more greatly affect coastal zones were added: “Off the Northeast Australian coast, the degradation of water quality, especially on reefs near the coast, soil erosion from beef grazing lands and fertilizer application to sugarcane cultivation [24-26].”

7. Line 60: Reference?

Response: We have added a reference to Fabricius, 2005 (doi: 10.1016/j.marpolbul.2004.11.028).

8. Line 69: Put this paragraph after first paragraph.

Response: Done.

9. Line 85: example of genera (e.g., Neogoniolithon)

Response: Done.

10. Line 157: Why did you put them out haphazardly? Cite what the research benefit may be and why you choose that method.

Response: “Haphazardly” was an incorrect word choice, thank you for catching this. Tiles had to be fixed onto hard non-living consolidated reef substratum, which was sometimes difficult to find. Therefore, tiles were placed within the range of 1 to 5 meters from one another

---

## [Editor Report · Decision Letter 1]

4 Feb 2025

Current velocity, water quality, and benthic taxa as predictors for coral recruitment rates on the Great Barrier Reef

PONE-D-24-35629R1

Dear Dr. Drake,

We’re pleased to inform you that your manuscript has been judged scientifically suitable for publication and will be formally accepted for publication once it meets all outstanding technical requirements.

Kind regards,

Tzen-Yuh Chiang

Academic Editor

PLOS ONE
---

## [Editor Report · Acceptance letter]

PONE-D-24-35629R1

PLOS ONE

Dear Dr. Drake,

I'm pleased to inform you that your manuscript has been deemed suitable for publication in PLOS ONE. Congratulations! Your manuscript is now being handed over to our production team.

Kind regards,

on behalf of

Dr. Tzen-Yuh Chiang

Academic Editor

PLOS ONE